# Comparison of Chayote (*Sechium edule* (Jacq.) Sw.) Accessions from Mexico, Japan, and Myanmar Using Reproductive Characters and Microsatellite Markers

**DOI:** 10.3390/plants12030476

**Published:** 2023-01-19

**Authors:** Miao Shi, Yihang Wang, Sergio Gabriel Olvera-Vazquez, Jorge Cadena Iñiguez, Min San Thein, Kazuo N. Watanabe

**Affiliations:** 1Graduate School of Science and Technology, University of Tsukuba, Tsukuba 305-8577, Japan; 2Graduate School of Life and Environmental Sciences, University of Tsukuba, Tsukuba 305-8577, Japan; 3GQE-Le Moulon, INRAe et University Paris-Saclay, Bâtiment Bréguet, 3 Rue Joliot Curie 2e ét, 91190 Gif-sur-Yvette, France; 4Colegio de Postgraduados, Campus San Luis Potosí, Salias de Hidalgo, San Luis Potosí 78622, Mexico; 5Department of Agricultural Research, Ministry of Agriculture, Livestock and Irrigation, Yezin, Myanmar; 6Tsukuba Plant Innovation Research Center, Institute of Life and Environmental Sciences, University of Tsukuba, Tsukuba 305-8577, Japan

**Keywords:** chayote, *Sechium edule*, cytology, palynology, morphology, microsatellite markers (SSR), population genetics

## Abstract

Promoting neglected and underutilized crop species is a possible solution to deal with the complex challenges of global food security. Chayote is a Neglected and Underutilized Cucurbit Species (NUCuS), which is recognized as a fruit vegetable in Latin America and is widely grown in Asia and Africa. However, basic biological knowledge about the crop is insufficient in scientific sources, especially outside of its center of origin. In this study, limited observations on reproductive characters were conducted, differentiating accessions from Mexico, Japan, and Myanmar. Cytological evaluation among Mexican and Japanese accessions showed that the relative nuclear DNA content is 1.55 ± 0.05 pg, the estimated genome size is 1511 at 2C/Mbp, and the observed mitotic chromosomal number is 2*n* = 28. The genetic diversity of 21 chayote accessions was also examined using six microsatellite markers. A global low genetic heterozygosity (*Ho* = 0.286 and *He =* 0.408) and three genetic groups were detected. The results established the basis to provide insights into chayote arrival history in Asia by looking at the crop’s reproductive morphology, cytology, and genetic diversity status outside its origin center. This could help in developing sustainable utilization and conservation programs for chayote.

## 1. Introduction

The biodiversity crisis began many years ago since human activities started to severely impact the global environment. The high rates of species extinction sped up the process of eroding the environment that human society depends on [1]. In the past 100 years, the crisis of losing genetic resources has been severe. Although humans have more than 12,000 options of edible crops in this world, less than 200 plant species have been fully utilized. Among these 200 species, maize, rice, and wheat are the source of over 60% of the energy consumed by humans daily. From this point of view, the demand to improve and promote the utilization of traditional and local species and cultivars has been raised [2,3].

One possible solution to deal with the complex challenges of global food security is to promote neglected and underutilized crop species. Given the vast repository of these crops, correct identification of the species with the potential to ease the stress of the food crisis around the world is another challenge for modern agriculture. In addition, the potential crops also need to withstand harsh environmental stresses and should have highly valuable market characteristics [4].

The utilization of cucurbits can be traced back more than 15,000 years [5]. The Cucurbitaceae family has been widely cultivated and plays a vital role in people’s daily diet, and has gradually become a significant part of the economy in tropical areas [5,6]. One of the characteristics of cucurbit fruits is high moisture and low fat; thus, they are regarded as an option for a healthy diet, which favors consumer interest [7]. Unlike the Cucurbit Popular Crops (CuPoC) comprising cucumber, pumpkin, squash, melon, and watermelon, chayote or *Sechium edule* (Jacq.) Sw. is considered a Neglected and Underutilized Cucurbit Species (NUCuS) [8].

Aside from the mature fruit of chayote, other parts of the plant such as the immature fruits, young leaves, shoots, and tuberous roots are also edible, providing nutrition such as fiber, protein, vitamins, unsaturated fatty acids, and lipids [9]. Hernandez-Uribe et al. (2011) reported that the Mexican chayote tubers are comparable with traditional starch sources such as potatoes in physicochemical and molecular characteristics, suggesting that it has the potential to be an alternative for starch production and commercialization [10].

It is suggested that the Mesoamerican region (mainly Mexico and Guatemala) could likely be the center of origin for chayote based on historical records [6,7,8,9,10,11]. After the conquest of Mexico, many crops of nutritional importance, including chayote, were transferred to different regions of the Spanish empire, starting the distribution outside their center of origin. Chayote was transferred to regions such as California, Louisiana, Hawaii, the Philippines, and Florida at the end of the 19th century [12]. In Japan, chayote was introduced from North America in 1917 [13].

For fertilization and seed development, pollen germination and pollen tube growth are prerequisites. Compared with investigating pollen physiologically and biochemically under in vivo conditions, in vitro pollen germination techniques are extensively used in various pollen systems [14]. In the process of hybridization, the pollen storage technique is considered to be the most efficient way to overcome the barriers in different flowering periods in other regions [15]. The exine pattern and apertural types of cucurbit pollen show significant variations wherein nearly all the shape classes have been reported in this family [16]. The authors also pointed out that these palynological characteristics are useful in identifying taxa at the generic level.

Cytological analyses are also essential for taxa identification and are a crucial prerequisite in breeding programs. A contrasting chromosomal number has been reported for *S. edule,* 2*n* = 22, 24, 26, and 28 [6,17,18,19,20]. This can be due to the small chromosome size and interference of secondary metabolites [21,22]. In addition, only one recent complete genome-level assembly has been reported in chayote and it is on the commercially-available variety [23]. Then, there is a need to study the ploidy level, chromosome number, and nuclear content of the different chayote varietal groups to obtain more accurate information.

Molecular markers have been proven as a powerful tool to analyze the genetic diversity of several species in Cucurbitaceae as another necessary requisite to develop breeding programs [24,25,26]. SSRs or microsatellites are stretches of DNA consisting of one or a few tandemly repeated nucleotides that have been widely applied as molecular markers since their discovery in the 1980s [27]. One study created specific SSR markers for chayote and reported a low level of genetic diversity among Mexican chayote accessions [28]. Chayote from other regions should also be studied to further evaluate genetic diversity.

This study aimed to detect the population structure and genetic diversity of chayote in Japan and Myanmar and compare it with that of Mexico, the center of origin. Information such as flower and fruit morphology, the basis of pollen germination, nuclear DNA content, and mitotic chromosome number were also provided. This information would help to better understand the current status of chayote in Asian regions and, in turn, build the foundation to develop sustainable programs for its protection and utilization.

## 2. Results

### 2.1. Palynology, Fruit, and Flower Morphology

Appendix A, as well as Table 1, show the morphological characteristics of flowers from the different *S. edule* accessions and varieties. It was noted that the number of petals of each variety ranged from 4 to 6, while the number of stamens ranged from 4 to 5.

As shown in Appendix A and Table 1, the pollen of one Japanese and five Mexican *S. edule* varieties presented diverse palynological characteristics under an optical microscope. Each one of the six varieties showed two types of pollen shape and the average thickness of the pollen wall ranged from 3.73 (KW019) to 8.18 (KW014). It is worth noting that, due to limitations in availability, the observations of the Mexican varieties were not comprehensive and therefore the data for this part may not be completely reliable.

Nine combinations of pH and maltose concentration were tested in this study, and the pollen germination results are shown in Figure 1a. The optimal pollen germination medium with pH 9.0 and 10% maltose produced the highest pollen germination rate of 70.6%. The germination rate using this optimal medium with different incubation times is shown in Figure 1b; it grows rapidly as the incubation time increases from 15 min to 60 min, then tends to be steady after 60 min. Therefore, the optimal pollen germination medium for *S. edule* is 10% Maltose (*w*/*v*), 0.25 mg/L H_3_BO_3_, and 1 mM Ca (NO_3_)_2_ at pH 9.0 with an incubation time more of than 60 min.

The results of the pollen longevity test are shown in Figure 2. At room temperature (25 °C), the pollen of both Mexican and Japanese varieties quickly lost all their germination potential. At a low temperature (4 °C), the reduction in the germinability of pollen was slower, and the Mexican *S. edule* varieties still had 7.08 to 10.50% germination potential even after seven days.

From the previous investigations [29,30,31], it was found that *S. edule* var. *virens levis* and *S. edule* var. *albus levis* were the major varieties in Myanmar. In this study, it was found that, in Japanese markets, these two varieties are also dominant. Therefore, the fruits of these two major varieties in Japan and Myanmar were collected and observed in this study. The fruit characteristics of five Mexican commercial varieties are described in Appendix A for comparison and the data came from a previous study using the same plant materials [8]. The fruit skins of the Mexican varieties *nigrum xalapensis* and *nigrum minor* were dark green, while that of *S. edule* var. *albus dulcis* was white-yellow. For the fruits from Japan and Myanmar varieties, the skin of *albus levis* was white or light green and smooth, while for *virens levis* it was only light green and tough (Appendix A).

### 2.2. Cytological Characterization

Using a flow cytometry experiment, the mean relative nuclear DNA content was 1.55 ± 0.05 pg; highest at 1.45 ± 0.06 pg and lowest at 1.61 ± 0.05 pg. The chayote varietal groups’ mean EGS was ~1511 2C/Mbp, ranging from ~1419 2C/Mbp to ~1573 2C/Mbp.

A mitotic chromosomal number of 2*n* = 28 was observed among the chayote accessions. For details, see Appendix A. The relative fluorescence intensity histogram results are shown in Appendix A.

### 2.3. Microsatellite-Based Genetic Diversity

Ten primer sets were tested, but only 6 showed clear amplification and polymorphism (Table 2). A total of 20 alleles were detected from these six primer sets, 3.33 on average for each locus. Effective alleles ranged from 1.477 (Sed07) to 2.116 (Sed03), with an average of 1.828. Shannon’s Information Index ranged from 0.394 (Sed07) to 0.737 (Sed09), with an average of 0.611. The lowest observed (*Ho*) and expected (*He*) heterozygosity were both found in Sed07 (0.000 and 0.270, respectively), while the highest *Ho* was in Sed06 (0.633), and the highest *He* in Sed09 (0.490). The average *Ho* and *He* were 0.286 and 0.408, respectively.

Both Wright’s Fst values and Nei’s genetic distances concluded similar results (Table 3). The biggest genetic differentiation was observed between Mexico and Myanmar groups (0.274), while the lowest was found between Mexico and Japan groups (0.137). In Nei’s genetic distance, the parameter for quantifying the degree of genetic divergence, the lowest value was observed between Japan and Mexico groups (0.303), which suggests a closer genetic relationship than the Myanmar group.

Bayesian clustering analysis was performed for the genetic structure of these 21 chayote accessions. Using structure harvester, the maximum delta *K* value was found at 3 (Appendix A); therefore, three hypothetical groups were identified. All Myanmar accessions were included in the first group, four Mexican and one Japanese in the second group, and seven Japanese and one Mexican in the third group (Figure 3). All Myanmar accessions had over 0.8 membership coefficient (*Q*) in the G1 genetic cluster; three Mexico accessions had around 0.8 membership coefficient in the G3 genetic cluster, while KW014 had 0.36 in G1 and KW015 had 0.41 in G2; six Japan accessions had over 0.8 in the G2 genetic cluster, KW030 had 0.39 mixed with G3, and KW019 even had around 0.9 membership coefficient in G3. The details of the membership coefficient (*Q*) are shown in Appendix A.

Principal coordinate analysis (PCoA) was carried out, with the first three axes of the PCoA explaining 81.34% of the variations, of which the first axis accounted for 39.92%, the second axis contributed 28.71%, and the third axis had 12.71% of the variations (Figure 4). The PCoA showed similar results as STRUCTURE: the accessions from the same origin were grouped together, with only two exceptions (KW015 and KW019). Japanese accession KW019 was closer to the Mexican group, and the Mexican accession KW015 was closer to the Japanese group. The original alleles of KNW11 and KNW12 are the same, and therefore cannot be separated in PCoA.

## 3. Discussion

Aung et al. (1990) stated that the *S. edule* flower has a calyx which is deeply five-partite and the five stamens have fused filaments [32]. The authors also indicated that the most suitable growing season of chayote is summer and early autumn months in temperate regions, but it can be adapted to a wide range of climatic conditions. In the review of Saade (1996), it is indicated that the wild type of *S. edule* has very similar (even identical in some cases) morphological characteristics to those of cultivated types. Additionally, the wild and cultivated types have the same staminal structure of flowers. Usually, the flower size of the wild types is slightly bigger than the cultivated ones. In addition, both petals and pedicels are five and triangular. Each petal is about 7 mm long and 2–3 mm wide [6].

In the current study, we observed that the stamen number is usually 4–5, while the petal number is usually 4–6. The petal width of all varieties ranged from 0.3 to 0.5 cm. The petal length of Mexican varieties ranged from 0.7 to 0.9 cm, which seems to be longer than the Japanese variety KW019 (0.3–0.5 cm). However, due to the biological nature of growing materials under foreign conditions, the survival rate and flowering phase of the varieties differ, making equal observation impossible. The observed differences between the Japanese and Mexican varieties may or may not be significant, but the flower morphology data of one representative Japanese chayote variety are valid and first reported here.

Ten varietal groups were classified based on morpho-structural, biochemical, physiological, and genetic diversity of chayote accessions from the *Sechium edule* National Germplasm Bank (BANGESe)—Autonomously University of Chapingo (UaCh) [33]. In our investigation, the major two chayote varieties in Myanmar and Japan are *virens levis* and *albus levis*. In these two regions, the var. *virens levis* and *albus levis* showed some different characteristics from the Mexican ones. Chayote has high diversity in fruit characteristics. For example, it has been reported that the var. *virens levis* in Costa Rica and Mexico, the first two worldwide exporters of chayote, have different fruit shapes. Costa Rican fruits are obovoid while Mexican fruits are piriform [12].

In our current and previous investigations [29,30,31], we observed that two varieties were dominant in Myanmar and Japan. They had three phenotypes: white smooth skin, light green tough skin, and light green smooth skin. Crossing is one possible explanation for these differences, which might have occurred during transferring and culturing. However, due to the lack of chayote-related history recording in these regions, it could be difficult to trace it back.

We detected a small genome size in *S. edule* (1511 2C/Mbp) compared with other reported cucurbit species. Our genome size values were like that reported for the sponge gourd 1525 2C/Mbp [34] but different to the snake gourd (2152 2C/Mbp) [35], or the pointed gourd (2220–5526 2C/Mbp [34,35,36,37]). We also confirmed that chayote is a diploid species with a chromosomal number of 28. This diploid level has been previously reported; but contrasting chromosomal numbers have been reported as 2*n* = 22, 23, 24, 26, and 28 [17,38,39,40,41,42]. A recent analysis of *S. edule* using Nanopore third-generation sequencing combined with Hi–C data estimated the genome size as 606.42 Mb and a chromosome number *n* = 14, which is consistent with our research [24]. One reason for the previously reported chromosomal number variation could be due to meiotic mutations that end in chromosomal translocations without genome losses or gains (dysploidy) [43].

Molecular markers represent powerful tools for genetic variation evaluation [44]. Microsatellites or SSR markers have been used to assess the genetic diversity in some species of the Cucurbitaceae family, such as cucumber, pumpkin, and bitter gourd germplasm [24,25,26]. Nevertheless, scarce studies have been utilized to assess genetic diversity in *S. edule*. For instance, Machida-Hirano et al. (2015) developed 10 SSR markers from a Mexican chayote collection, which were used in our study [28]; we detected relatively low genetic diversity and heterozygosity, which is consistent with that study. Self-compatibility, the utilization of a few plants to obtain fruits, and backyard cultivations could explain the relatively low diversity [12,13,14,15,16,17,18,19,20,21,22,23,24,25,26,27,28,29,30,31,32,33,34,35,36,37,38,39,40,41,42,43,44,45,46]. However, our results contrast with reports using different molecular markers with high genetic diversity, such as isozyme in Costa Rican accessions [46], and moderate-to-high genetic diversity analyzing Indian germplasm accession using RAPD and ISSR markers [47]. These contrasting results could be due to the nature of the molecular marker. For instance, isoenzymes are not recommended because the environment regulates their expression [43], the low specificity of the genome of ISSR primers [48], the dominant nature of ISSR and RAPD markers that has limited extracting heterozygosity information [49] as well as the lack of reproducibility, which has been reported in RAPD and ISSRs [50]. Low genetic differentiation but high morphological variation has been reported in chayote. The environment (e.g., temperature, light, humidity, and precipitation) and human practices (e.g., traditional knowledge and production system) are important sources of variation [51].

We detected three main groups that represent their origin by evaluating the genetic structure using STRUCTURE and PCoA. We saw a defined Myanmar group (G1) and two more groups showing some shared ancestry between Japanese and Mexican accessions. The Japanese accession KW019 was closer to the Mexican group (G3), and the Mexican accession KW015 was more relative to the Japanese group (G2). The chayote was transferred to different regions of the Spanish empire from Mesoamerica, the chayote’s center of origin [12,13,14,15,16,17,18,19,20,21,22,23,24,25,26,27,28,29,30,31,32,33,34,35,36,37,38,39,40].

The chayote was introduced into Japan from North America in 1917 and was called “hayatouri” [13]. Thus, these findings suggest that, even though this crop was introduced in Japan around 100 hundred years ago, seasonal and backyard production has a low impact on the chayote genetic variation [33]. However, we should increase the number of samples to obtain more reliable results.

Both the ancestry analysis and genetic differentiation statistics support that the Myanmar group is relatively independent from the Japan and Mexico groups. This may indicate that the Myanmar group was transferred to this area from a different route, which could probably be through Europe to Africa and then Myanmar or through the Philippines to Myanmar; the diversity was gradually lost during the transfer; only a few accessions arrived and acclimatized. The local crop varieties in Myanmar can be influenced by neighboring India and/or China, or vice versa. Since chayote is distributed in both India [47] and southwest China [23], this influence on chayote might be enhanced. Limited information is available about the history of the introduction of chayote in Myanmar. Therefore, new research is open to disentangle the arrival of chayote in Myanmar and its neighboring countries through historical records and molecular tools. Our findings allowed us to detect a constant diploid chromosome number but low levels of genetic diversity and evaluate chayote accessions from different origins. We also suggested possible routes that chayote was transferred into Japan and Myanmar. However, it is necessary to increase the number of samples, expand the geographic sampling range, and utilize more informative markers (i.e., Single Nucleotide Polymorphisms, SNPs) to assess the chayote genetic diversity and morphological variation. Our study also established preliminary information to develop research to understand the response of chayote under different environments and management (i.e., commercial, cultural, and conservation) [52].

In conclusion, our study focused on chayote in Japan and Myanmar, taking Mexican accessions for comparison. We provided morphological, cytological, and genetic diversity information which could help us understand the transfer history, and current situation of chayote in the Asian area, and serve for future breeding and conservation programs (i.e., hybridization and in vitro fertilization).

## 4. Materials and Methods

### 4.1. Plant Material

To provide a holistic study of chayote, different plant materials and varieties were collected from Mexico, Japan, and Myanmar. Details on these are provided in Appendix A.

In vitro cultures of five Mexican commercial *S. edule* varieties were provided by the Interdisciplinary Mexican Research Group of *Sechium edule* (GISeM)/National Genetics Resources Center (CNRG). The materials have an Internationally Recognized Certificate of Compliance (IRCC), ABSCH-IRCC-MX-208823-1, under the Mexican national legal regimes associated with the Nagoya Protocol on Access and benefit-sharing of Genetic Resources for the Convention on Biological Diversity, initiated in 2017. These cultures were maintained in the Gene Research Center (GRC) of the Tsukuba Plant Innovation Research Center, University of Tsukuba, Japan through the Science and Technology Research Partnership for Sustainable Development (SATREPS) program.

Nine Japanese accessions are representative varieties collected from local markets in Tsukuba, Chiba, and Kagoshima.

The Mexican and Japanese accessions were grown up to maturity in a greenhouse at GRC under the same conditions.

Moreover, Myanmar supplied the dried leaf samples through the Seed Bank of the Department of Agricultural Research (DAR), Ministry of Agriculture, Livestock, and Irrigation with a Material Transfer Agreement in 2019 under a Memorandum of Understanding (MoU) and its specific research agreement on the sample collection under Mutually Agreed Terms. The samples were kept dry in indicating silica gel prior to their use. In addition, fruit samples were also collected from the local markets and farmers’ yards.

This study utilized a total of 22 chayote accessions wherein each part had to use different accession and/or materials depending on their availability as well as purpose.

### 4.2. Palynology, Fruit, and Flower Morphology

#### 4.2.1. Flower and Pollen Observation

One Japanese accession (KW019), along with five Mexican accessions (KW014, KW015, KW016, KW017, and KW018) were used for this part (Table 1).

In 2020, the Japanese variety formed its flower from the beginning of September up to the beginning of November, while that of the Mexican varieties formed from mid-November to the near end of December. For Japanese variety, 50 flowers were taken and observed; for Mexico varieties, unfortunately, due to the biological nature of growing in foreign conditions, only limited number of flowers were observed. The flowers were taken from the field at 9–10 AM, then observed and described based on the same morphological descriptors used by Saade (1996) [6].

Pollen was also collected from the anthers following the method used by Qureshi et al. (2009) [53]. Mature anthers were crushed and pollen grains were mixed thoroughly with a drop of 1% aceto-carmine staining solution placed on a glass slide, then gently covered with a cover slip. Three slides were prepared for each flower. Slides were observed under an optical microscope at ten randomly selected fields under the 10 × objective (100 × total magnification). A total of 300 pollen grains were counted in each cultivar and 3 repetitions were carried out for each cultivar.

#### 4.2.2. Pollen Germination Medium Optimization

One representative Japanese variety *virens levis* (KW019) was used for this part. Anthers were taken at 9–10 AM from October to December 2019, then crushed and kept in airtight small plastic container with high humidity in dark for incubation. Basic germination medium follows the protocol as Vižintin and Bohanec (2004) described [54], consisting of 15% sucrose (*w*/*v*), 0.25 mg/L H_3_BO_3_, and 1 mM Ca (NO_3_)_2_ at pH 7.0. After 60 min incubation, 500–600 pollen grains per treatment were examined under an optical microscope with 3 repetitions. Pollen is considered germinated when pollen tube equaled or exceeded the diameter of the pollen grain. In this study, sucrose was replaced by maltose, and 9 different combinations of pH (7.0, 8.0, 9.0) and maltose concentration (10%, 12.5%, 15%) were tested with an incubation time of 60 min to obtain the optimal combination. Then, the optimal medium was used for pollen germination with different incubation time changes from 15 to 180 min.

#### 4.2.3. Pollen Longevity Test

The optimized germination medium was used to check the longevity of pollen from the described flowers of the Japanese and Mexican chayote accessions. It was checked after 1, 2, 3, and 7 days of storage at the temperatures 4 and 25 °C. The same method for slide preparation and observation, described in Section 4.2.2, was used to determine the percent germinability of the pollen.

#### 4.2.4. Fruit Observation

Mature fruits of the two dominant varieties (*S. edule* var. *virens levis* and *S. edule* var. *albus levis*) in Myanmar and Japan were collected from the local markets and farmers’ yards. Photographs were taken to show the approximate morphological features (Appendix A). The observation methods used were from previous studies [55,56,57].

### 4.3. Cytological Characterization

#### 4.3.1. Experiment Reliability Performance

Two Japanese and five Mexican chayote accessions were used for this part. Experiment reproducibility was assessed through an Analysis of Variance (ANOVA) using the measurement of three repetitions of each accession obtained on different days in the same time period and under equal conditions. The reported tetraploid potato (*Solanum tuberosum* L.) was utilized as an external standard nuclear DNA content (2C-value) because its 2C-value was previously reported as 3.2 pg [35]. In addition, the limit of the coefficient of variation was 5.0%, evaluated between 5000 to 10,000 cells.

#### 4.3.2. Flow Cytometry Experiment

The experiment was performed using flow cytometry equipment (Partec^®^) following the Cystain^®^ UV Precise P kit (*Partec GmbH*, Münster, Germany) instructions. The relative 2C-value was computed based on the following formula [58]:$$\mathrm{Sample}\;2C\;\mathrm{value}=\mathrm{Reference}\;2C\;\mathrm{value}\;\text{×}\;\frac{\mathrm{sample}\;2C\;\mathrm{mean}\;\mathrm{peak}\;\mathrm{position}}{\mathrm{reference}\;2C\;\mathrm{mean}\;\mathrm{peak}\;\mathrm{position}}$$

The estimated genome size (EGS) was calculated using 1 pg = 978 Mbp [59]. Significant differences were assessed using ANOVA. Tukey’s test was utilized to address the relative 2C-values of species and among accessions (*p* < 0.05).

#### 4.3.3. Chromosomal Counting

The root tips were treated using the following procedure: pretreated in 8-hydroxyquinoline for 4 h at 18 °C, Carnoy’s solution (60% ethanol, 30% Chloroform, and 10% glacial acetic acid) fixation step for 24–48 h, a maceration in 1N HCl for 5 min at 55 °C and stored in 70% aqueous ethanol. The slides were prepared using standard squash technique and aceto-carmine staining. The observations were made with light microscope, Olympus BX53 (Olympus Corporation, Japan).

### 4.4. Microsatellite-Based Genetic Diversity

#### 4.4.1. DNA Isolation

Fruit skin material of 8 Myanmar accessions was dehydrated with silica gel and kept dry until DNA isolation. For 5 Mexico and 8 Japan accessions, young leaf material was sampled and kept at −80 °C. DNA was isolated using modified CTAB method [60]. DNA completeness was examined by 2% agarose gel electrophoresis at 100 V, 15 mA for 25 min. Quality and purity of DNA was assessed by NanoDrop 2000c spectrophotometer.

#### 4.4.2. SSR Genotyping

From the ten SSR primer sets by Machida-Hirano [28], six showed clear amplification and polymorphism and were used for this research (Table 2). The PCR reaction mixture was 10 μL in total, consisting of 0.1 μL Takara ExTaq (5U/μL), 1 μL 10× ExTaq buffer, 0.8 μL dNTP mixture, 1 μL each of forward and reverse primer, 2 μL DNA sample (20 ng/μL), and 4.1 μL double distilled water. PCR reactions were carried out in GeneAmp PCR System 9700 (Thermo Fisher, Waltham, Massachusetts, USA). The PCR program was set as: initial denaturation at 95 °C for 3 min, followed by 30 cycles of denaturation at 95 °C for 30 s, primer annealing at 60 °C for 30 s and an extension at 72 °C for 1 min, then final extension at 72 °C for 60 min. The PCR product was then mixed with Midori Green Advance DNA stain (Nippon Genetics, Tokyo, Japan) and Sample Treatment for Tris Acid (Nacalai Tesque, Kyoto, Japan) following the given instructions. A total 10 μL of mixture was added in one well of 10% polyacrylamide gel.

Polyacrylamide gel electrophoresis (PAGE) was carried out at 100V, 15mA in 1x TBE buffer for 2 h. Thereafter, the gel was scanned and bands were identified with Gel Doc XR Imaging System (Bio-Rad, Hercules, CA, USA). An example of results is shown in Appendix A.

#### 4.4.3. Data Analysis

The co-dominant data were recorded in Microsoft Excel^®^ and processed with a plug-in named GeneAlEx 6.503 to calculate Shannon’s information index (*I*), number of observed alleles (*Na*), number of effective alleles (*Ne*), expected heterozygosity (*He*), and observed heterozygosity (*Ho*); Wright’s Fst values (Fst) and Nei’s genetic distances (D) were also calculated [61,62]. STRUCTURE 2.3.4 was used to estimate the number of genetic clusters with the length of burn-in period of 50,000 and 100,000 of MCMC (Markov Chain Monte Carlo) repetitions after burn-in; *K* was set from 1 to 10 with 10 iterations [63,64,65,66]. The results were uploaded to Structure Harvester to calculate the optimum value of population clusters (*K*) using Evanno’s method, which detects the strongest level of population subdivision [67,68]. All 10 replicates from the best *K* were aligned using CLUMPP 1.1.2 [69]. Hierarchical cluster analysis and a percent stacked chart of hypothetical ancestry (*Q)* were plotted with R package ggplot [70] and ggtree [71,72,73]. Principal coordinate analysis (PCoA) was performed and plotted with GeneAlEx 6.503 [61,62].

## Figures and Tables

**Figure 1 plants-12-00476-f001:**
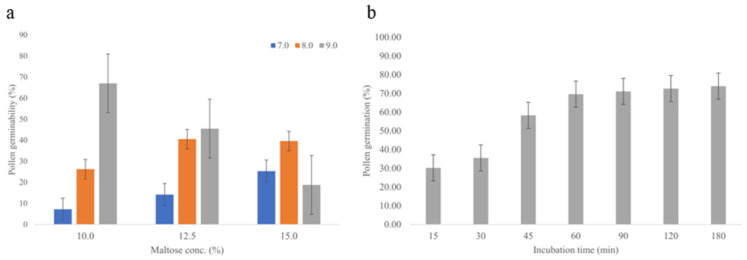
The germination rate of pollen (**a**) in medium with various pH values and maltose concentration combinations after 60 min incubation and (**b**) in the optimized medium under different incubation periods. Error bars indicate ± S.E.

**Figure 2 plants-12-00476-f002:**
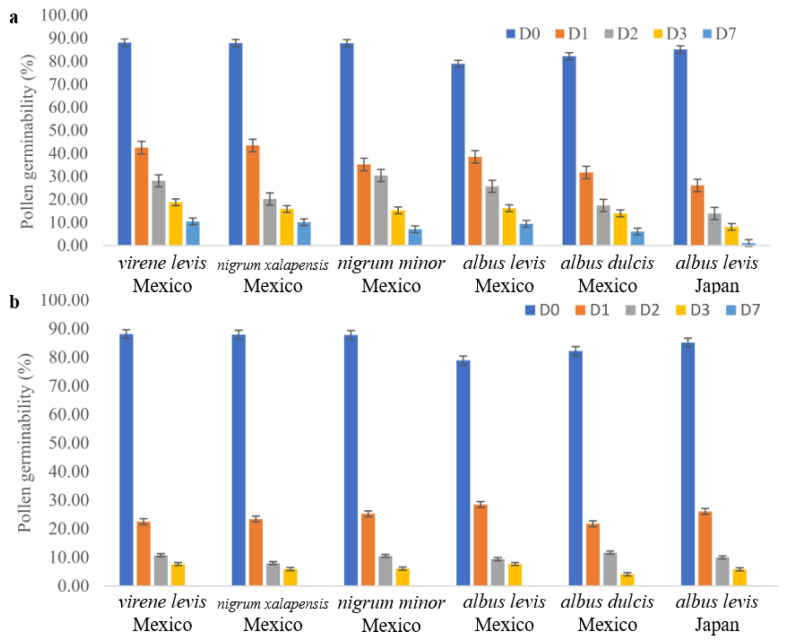
Pollen viability of 5 Mexican and 1 Japanese *S. edule* after 7 days. Error bars indicate ± S.E. (**a**) under 4 °C. (**b**) under 25 °C.

**Figure 3 plants-12-00476-f003:**
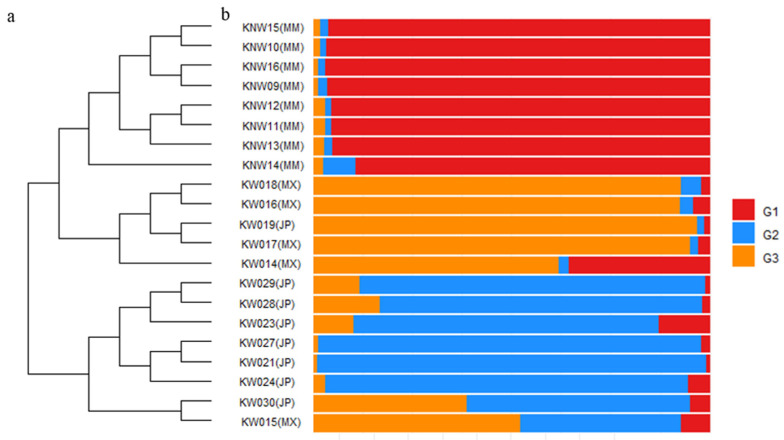
(**a**) Hierarchical cluster analysis based on hypothetical ancestry (*Q*) using Lance–Williams algorithms. The parenthesis acronyms refer to each accession’s origin, MM = Myanmar, MX = Mexico, JP = Japan (**b**) STRUCTURE analysis of 21 chayote accessions. G1, G2, and G3 refer to 3 hypothetical ancestries (*Q*) in the genetic clusters.

**Figure 4 plants-12-00476-f004:**
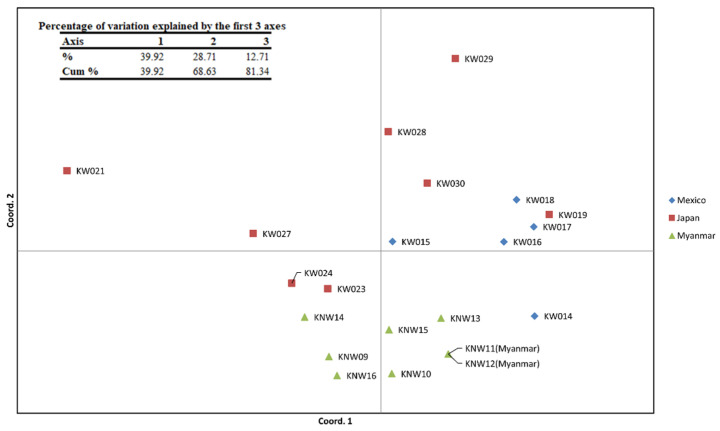
The principal coordinate analysis (PCoA) of twenty-one *S. edule* accessions based on the SSR data. The first three axes of PCoA explained 81.34% of variations.

**Table 1 plants-12-00476-t001:** The comparative flower morphological and palynological characteristics of 1 Japanese and 5 Mexican *S. edule* varieties.

			Flower Morphological Characteristics					Palynological Characteristics				
Accession	Variety Name	Origin	Colour	Number of Petals	Petal Length (cm)	Petal Width (cm)	Number of Stamens	Polar Axis (µm)	Equatorial Axis (µm)	P/E Ratio	Shape	Pollen Wall (µm)
KW014	*virens levis*	Mexico	Light green	4, 5	0.7–0.9	0.3–0.5	5	100.16	85.76	1.17	Subprolate	8.18
								98.40	99.31	0.99	Oblate-spheroidal	
KW015	*nigrum xalapensis*	Mexico	Yellowish	5, 6	0.7–0.9	0.3–0.5	4, 5	97.06	86.12	1.13	Prolate-spheroidal	4.70
								95.54	94.26	1.01	Oblate-spheroidal	
KW016	*nigrum minor*	Mexico	Dark green	5	0.7–0.8	0.3–0.5	4, 5	92.71	85.35	1.09	Prolate-spheroidal	5.47
								85.75	82.68	1.04	Oblate-spheroidal	
KW017	*albus levis*	Mexico	Yellowish	5, 6	0.7–0.9	0.3–0.5	5	85.65	75.58	1.13	Subprolate	5.96
								84.29	82.44	1.02	Oblate-spheroidal	
KW018	*albus dulcis*	Mexico	Yellowish	5	0.8–0.9	0.3–0.5	5	94.58	83.48	1.13	Subprolate	6.14
								96.17	93.62	1.03	Oblate-spheroidal	
KW019	*albus levis*	Japan	White	4, 5	0.3–0.5	0.3–0.4	4, 5	104.83	88.94	1.18	Subprolate	3.73
								103.80	102.69	1.01	Oblate-spheroidal	

**Table 2 plants-12-00476-t002:** The estimated polymorphic information on chayote using SSR markers.

Locus	*N*	*Na*	*Ne*	*I*	*Ho*	*He*	*uHe*	*F*
Sed11	7.000	3	1.535	0.467	0.217	0.284	0.307	0.140
Sed09	7.000	3	1.960	0.737	0.500	0.490	0.530	−0.014
Sed08	7.000	3	1.905	0.692	0.125	0.458	0.494	0.758
Sed07	6.333	3	1.477	0.394	0.000	0.270	0.294	1.000
Sed06	7.000	3	1.977	0.723	0.633	0.471	0.507	−0.364
Sed03	7.000	5	2.116	0.654	0.238	0.477	0.382	0.122
Average	6.889	3.333	1.828	0.611	0.286	0.408	0.419	0.274

*N* = Number of samples, *Na* = Number of Observed Different Alleles, *Ne* = Number of Effective Alleles, *I* = Shannon’s Information Index, *Ho* = Observed Heterozygosity, *He* = Expected Heterozygosity, *uHe* = Unbiased Expected Heterozygosity, *F* = Fixation Index.

**Table 3 plants-12-00476-t003:** Pairwise Population Matrix of Wright’s Fst Values and Nei’s Genetic Distance.

Fst	Japan	Mexico	Myanmar	D	Japan	Mexico	Myanmar
Japan	0.000			Japan	0.000		
Mexico	0.137	0.000		Mexico	0.303	0.000	
Myanmar	0.199	0.274	0.000	Myanmar	0.422	0.440	0.000

Fst = Wright’s Fst values, on the left; D = Nei’s Genetic Distances, on the right.

## Data Availability

Not applicable.

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
