# Peer review of "Comparison of Chayote (*Sechium edule* (Jacq.) Sw.) Accessions from Mexico, Japan, and Myanmar Using Reproductive Characters and Microsatellite Markers"

_plants, 2023, doi:10.3390/plants12030476_

Round 1

Reviewer 1 Report

1. This manuscript has three purposes. However, it is recognized that the abstract and manuscript do not incorporate the three purposes. In addition, comprehensive conclusions from these three experiments are needed. Although it is a single study, the analysis of morphology, cytology, and diversity are recognized as separate contents. It is recommended to add conclusions at the end of the discussion and abstract.

2. Among the SSRs developed in Machida-Hirano's research, it was written that 10 polymorphisms and amplifications were possible. Why did you use 6 of these SSRs? Please write the conditions for selection of 6 markers in the manuscript.

3. Using 6 SSRs, the diversity of all resources used in this study was confirmed (Table 2). Is it possible to analyze the diversity of each country's resources? It is thought to improve the quality of this study.

Author Response

Response to Reviewer 1 Comments and Suggestions

Thank you very much for your valuable comments and suggestions, they have been very helpful in improving our manuscript. We have revised the manuscript according to the comments and suggestions, and have also invited a native speaker to help with the English editing.

Here is our response to each of them:

Point 1:

This manuscript has three purposes. However, it is recognized that the abstract and manuscript do not incorporate the three purposes. In addition, comprehensive conclusions from these three experiments are needed. Although it is a single study, the analysis of morphology, cytology, and diversity are recognized as separate contents. It is recommended to add conclusions at the end of the discussion and abstract.

Response 1:

Chayote was introduced into Asia for hundreds of years. However, scientific information is very limited. Our study is focused on the chayote outside its origin center, taking Mexican accession as comparison. We mainly research on the genetic diversity, and also reported some morphology and cytology information.

To emphasize this, we altered some statements and added conclusions in Abstract, Introduction and Discussion.

Point 2:

Among the SSRs developed in Machida-Hirano's research, it was written that 10 polymorphisms and amplifications were possible. Why did you use 6 of these SSRs? Please write the conditions for selection of 6 markers in the manuscript.

Response 2:

In our pre-experiment, one of those markers didn’t produce polymorphic bands, three didn’t produce bands with expected size. Therefore, we only used the six SSRs which are polymorphism and clear for current study.

Point 3:

Using 6 SSRs, the diversity of all resources used in this study was confirmed (Table 2). Is it possible to analyze the diversity of each country's resources? It is thought to improve the quality of this study.

Response 3:

We added Table 3 and some explanation for it, which provides the pairwise population matrix of Wrigh’s Fst values and Nei’s genetic distances between the three groups from different countries. These statistics can help us understand the genetic differentiation of each country’s resources.

Table 3 Pairwise Population Matrix of Wrigh’s Fst Values and Nei’s Genetic Distance.

Fst

Japan

Mexico

Myanmar

D

Japan

Mexico

Myanmar

Japan

0.000

Japan

0.000

Mexico

0.137

0.000

Mexico

0.303

0.000

Myanmar

0.199

0.274

0.000

Myanmar

0.422

0.440

0.000

Fst = Fst

Reviewer 2 Report

The manuscript is very confusing.

1. It does not clearly identify a research problem

2. Authors declare in the Abstract and the Introduction the use of certain genetic materials; however, in the materials and methods section the varieties used change radically between the different sections, to such a degree that they cause much confusion. The dendrogram of Figure 3 shows 21 accessions!! The authors should prepare a table showing all the accessions studied and mark those that were involved in each section of study.

3. The use of different varieties for each section of the study is very heterogeneous that it is difficult to achieve a real integration of the information so that it can be useful for the design of conservation programs for this species.

4. The existence of statistical differences for the different characters of the morphological description of the flower is not shown (Table 1), in the case of petal characteristics, the average is not even shown (only ranges are shown), so it is not possible to draw valid conclusions.

6. The fruits were taken only from two different varieties directly from orchards, so the lack of representativeness for a scientific manuscript is evident; In addition, the information was derived from photographs of them.

7. It is mentioned that an analysis of variance was applied but its results are not shown, nor is the existence of statistical significance even mentioned throughout the manusript.

8. On repeated occasions it is mentioned that there is reduced genetic differentiation; but the statement is not supported by any relevant statistics (e.g. Wright´s Fst).

Author Response

Response to Reviewer 2 Comments and Suggestions

Thank you very much for your valuable comments and suggestions, they have been very helpful in improving our manuscript. We have revised the manuscript according to the comments and suggestions, and have also invited a native speaker to help with the English editing.

Here is our response to each of them:

Point 1:

It does not clearly identify a research problem

Response 1:

Chayote was introduced into Asia for hundreds of years. However, scientific information is very limited. Our study is focused on the chayote outside its origin center, taking Mexican accession as comparison. We mainly research on the genetic diversity, and also reported some morphology and cytology information.

To emphasize this, we altered some statements and added conclusions in Abstract, Introduction and Discussion.

Point 2:

Authors declare in the Abstract and the Introduction the use of certain genetic materials; however, in the materials and methods section the varieties used change radically between the different sections, to such a degree that they cause much confusion. The dendrogram of Figure 3 shows 21 accessions!! The authors should prepare a table showing all the accessions studied and mark those that were involved in each section of study.

Response 2:

This might have been confused on our initial statement.

For genetic study, we have 21 accessions from 3 regions; for morphology and cytology observation we mainly used Japanese and Mexican accessions. This study utilized a total of 22 chayote accessions wherein each part used different accession and/or materials depending on their availability as well as purpose.

In Table S4, we have provided information on the plant materials used in this study, including the code, culture condition, variety name, source and sections where they were used.  Additionally, we have included additional text descriptions in each section to further clarify.

Point 3:

The use of different varieties for each section of the study is very heterogeneous that it is difficult to achieve a real integration of the information so that it can be useful for the design of conservation programs for this species.

Response 3:

Thank you for noticing this issue.

Our main goal is to assess the genetic diversity of chayote outside its origin center, and also provide other biological information (morphology and cytology). So, we have emphasized the genetics portion more and only simply reported the morphology and cytology.

Point 4:

The existence of statistical differences for the different characters of the morphological description of the flower is not shown (Table 1), in the case of petal characteristics, the average is not even shown (only ranges are shown), so it is not possible to draw valid conclusions.

Response 4:

We are not aimed to show the differences between the two groups. We referred to previous reports on chayote floral morphology and observed the petals and stamens. In those studies, the characteristics were reported as ranges or integer. (Discussion paragraph 1)

Point 6:

The fruits were taken only from two different varieties directly from orchards, so the lack of representativeness for a scientific manuscript is evident; In addition, the information was derived from photographs of them.

Response 6:

We might not explain well.

We investigated many states in Myanmar and prefectures in Japan, not only in orchard but also in market and wild, and found out there are two dominant varieties (albus levis has two phenotypes, virens levis has one) in these two countries. Therefore, we described and took photo of the fruits from these two varieties. Now this text description is added to the article, and also cited the investigation reports.

Point 7:

It is mentioned that an analysis of variance was applied but its results are not shown, nor is the existence of statistical significance even mentioned throughout the manuscript.

Response 7:

Due to the biological nature to grow materials under Tsukuba conditions (foreign climate, and survival rate in acclimatization from in vitro), it’s hard to observe Mexican and Japanese accessions equally. We can’t collect enough data from Mexican accessions for ANOVA to show significant differences between Mexico and Japanese accessions; but the observation of Japanese accessions is valid, so we decide to report it since we don’t talk about the morphology differences between these two groups in our discussions now, and this is the first report on Japanese chayote.

We added explanation for this in Results and Materials & Methods.

Point 8:

On repeated occasions it is mentioned that there is reduced genetic differentiation; but the statement is not supported by any relevant statistics (e.g. Wright´s Fst).

Response 8:

We added Table 3 and some explanation for it, which provides the pairwise population matrix of Wrigh’s Fst values and Nei’s genetic distances between the three groups from different countries. This can support the statement that chayote has a low genetic differentiation among groups, and Myanmar accessions are relatively independent from Japan and Mexico ones.

Table 3 Pairwise Population Matrix of Wrigh’s Fst Values and Nei’s Genetic Distance.

Fst

Japan

Mexico

Myanmar

D

Japan

Mexico

Myanmar

Japan

0.000

Japan

0.000

Mexico

0.137

0.000

Mexico

0.303

0.000

Myanmar

0.199

0.274

0.000

Myanmar

0.422

0.440

0.000

Fst = Fst values, on the lef

Round 2

Reviewer 2 Report

1. The new version offers a better justification for conducting the study. A few details remaining:

2. Lines 38-40.  Introduction. The information provided by the authors in lines 38-40 is appropriate for other crops such as maize, wheat and rice; however, it is not the situation that prevails in chayote, since the use of improved varieties is not so common in this species.

3. The values of both observed and expected heterozygosity are actually not as low as claimed by the authors.

3. Lines 431-432. Materials and Methods. Authors wrote: from the beginning of November up to the beginning of September. Probably they mean "from the beginning of September up to the beginning of November"

4. Results. Table 1. If authors performed analysis of variance to the data of floral morphology and pollen traits, they should report in Table 1 the mean values (in addition to the ranges) and indicate the existence (or absence) of statistical significance between the accessions.

4, Results. Line 143, Table 2. The fifth column (PIC) is unnecessary, since it refers to values of the diversity revealed by the markers, rather than of the accessions directly.

5. Bottom of table 2. Clearly explain the difference between Na and Ne.

6. Results, Lines 168-170. Authors should be careful not to confuse the concepts of genetic differentiation (Wright's Fst Statistic) with genetic divergence (Nei's Genetic Distance).

Author Response

Thank you very much for your kind and valuable comments and suggestions, we have improved our manuscript according to that.

Here is our response to each of them:

  1. Lines 38-40. Introduction. The information provided by the authors in lines 38-40 is appropriate for other crops such as maize, wheat and rice; however, it is not the situation that prevails in chayote, since the use of improved varieties is not so common in this species.

Response:

That description has been deleted.

  1. The values of both observed and expected heterozygosity are actually not as low as claimed by

the authors.

Response:

This statement is based on the comparison with other cucurbit species.

In current study: Ho = 0.286, He = 0.408; in melon (Cucumis meloL.)1: Ho = 0.52, He = 0.61; in bitter gourd (Momordica charantia Linn.)2: Ho = 0.621. In the article where the SSR markers for chayote was published, the discussion section says:

Other genetic diversity studies on Cucurbitaceae using microsatellite markers demonstrated moderate to high heterozygosity (0.49 - 0.75 [28] , 0.26 - 0.79 [29] , 0.00 - 1.00 [30] , and 0.325 - 0.867 [31] ).

  1. Lines 431-432. Materials and Methods. Authors wrote: from the beginning of November up to the beginning of September. Probably they mean "from the beginning of September up to the beginning of November"

Response:

Thanks for reminding, it has been corrected.

  1. Results. T able 1. If authors performed analysis of variance to the data of floral morphology and pollen traits, they should report in Table 1 the mean values (in addition to the ranges) and indicate the existence (or absence) of statistical significance between the accessions.

Response:

We intended to provide some reference info rather as supplementary information since our main goal is to show the presence of the genetic diversity rather than morphology differences.

4, Results. Line 143, T able 2. The fifth column (PIC) is unnecessary, since it refers to values of the

diversity revealed by the markers, rather than of the accessions directly.

Response:

That column and PIC relevant descriptions has been deleted.

  1. Bottom of table 2. Clearly explain the difference between Na and Ne.

Response: Yes, the explanation is added.

Na is the observed different alleles. Ne, the effective number of alleles, is the number of alleles with the same frequency that are needed in a population to achieve the same expected heterozygosity, which is lower than the observed different alleles (Na) in general.

Ne = 1 / (Sum pi^2), where pi is the frequency of the ith allele for the population & sum pi^2 is the sum of the squared population allele frequencies.

  1. Results, Lines 168-170. Authors should be careful not to confuse the concepts of genetic differentiation (Wright's Fst Statistic) with genetic divergence (Nei's Genetic Distance).

Response:

Sorry for confusing. We added extra explanation in that paragraph to clarify it.

References:

(1)          Ritschel, P. S.; Lins, T. C. de L.; Tristan, R. L.; Buso, G. S. C.; Buso, J. A.; Ferreira, M. E. Development of Microsatellite Markers from an Enriched Genomic Library for Genetic Analysis of Melon (Cucumis MeloL.). BMC Plant Biology2004, 4 (1), 1–14.

(2)          Ji, Y.; Luo, Y.; Hou, B.; Wang, W.; Zhao, J.; Yang, L.; Xue, Q.; Ding, X. Development of Polymorphic Microsatellite Loci in Momordica Charantia (Cucurbitaceae) and Their Transferability to Other Cucurbit Species. Scientia horticulturae 2012,140, 115–118.

Over.